# First Principles Study of p-Type Transition and Enhanced Optoelectronic Properties of g-ZnO Based on Diverse Doping Strategies

**DOI:** 10.3390/nano14231863

**Published:** 2024-11-21

**Authors:** Kaiqi Bao, Yanfang Zhao, Wei Ding, Yuanbin Xiao, Bing Yang

**Affiliations:** 1School of Mechanical Engineering, Jiangsu University of Technology, Changzhou 213001, China; 2022655174@smail.jsut.edu.cn (K.B.); xiaoyuanbin@jsut.edu.cn (Y.X.); 2School of Electrical Engineering and Automation, Changshu Institute of Technology, Changshu 215500, China; dingwei@cslg.edu.cn; 3Key Laboratory of Materials Surface Science and Technology, Jiangsu Province Higher Education Institutes (Changzhou University), Changzhou 213164, China; 4Centre for Advanced Laser Manufacturing (CALM), School of Mechanical Engineering, Shandong University of Technology, Zibo 255000, China; yangbingem@sdut.edu.cn

**Keywords:** first principles, p-type, g-ZnO, optical properties

## Abstract

By utilizing first principles calculations, p-type transition in graphene-like zinc oxide (g-ZnO) through elemental doping was achieved, and the influence of different doping strategies on the electronic structure, energy band structure, and optoelectronic properties of g-ZnO was investigated. This research study delves into the effects of strategies such as single-acceptor doping, double-acceptor co-doping, and donor–acceptor co-doping on the properties of g-ZnO. This study found that single-acceptor doping with Li and Ag elements can form shallow acceptor levels, thereby facilitating p-type conductivity. Furthermore, the introduction of the donor element F can compensate for the deep acceptor levels formed by double-acceptor co-doping, transforming them into shallow acceptor levels and modulating the energy band structure. The co-doping strategy involving double-acceptor elements and a donor element further optimizes the properties of g-ZnO, such as reducing the bandgap and enhancing carrier mobility. Additionally, in terms of optical properties, g-Zn_14_Li_2_FO_15_ demonstrates outstanding performance in the visible-light region compared with other doping systems, especially generating a higher absorption peak around the wavelength of 520 nm. These findings provide a theoretical foundation for the application of g-ZnO in optoelectronic devices.

## 1. Introduction

ZnO is a direct-bandgap semiconductor material belonging to Groups II-VI, together with GaN and SiC, which constitute the third-generation semiconductor materials. At room temperature, ZnO has a bandgap of 3.37 eV, an exciton binding energy of up to 60 meV, a relatively low dielectric constant, and excellent photoelectric properties [1,2,3]. To better realize the device applications of ZnO, it is necessary to develop reliable n-type ZnO and p-type ZnO. Due to the zinc interstitials and oxygen vacancies in ZnO, these intrinsic defects endow ZnO with weak n-type conductivity, facilitating research on n-type ZnO [4,5,6]. In contrast, the pursuit of stable and reliable p-type ZnO is more challenging. Currently, the p-type transition of ZnO is primarily achieved through the mono-doping or co-doping of ZnO with acceptor elements from Group I (Li, Na, Ag, Cu, etc.) and Group V (N, P, As, etc.) [7,8,9,10]. Moreover, the realization of p-type ZnO has also been explored via the co-doping of acceptor and donor elements (Al-N, In-N, Ga-N, etc.) [11,12,13].

The discovery of graphene in 2004, with its remarkable properties, like high electron mobility, mechanical strength, and thermal conductivity, paved the way for the exploration of two-dimensional (2D) structures [14,15,16,17]. Two-dimensional nanomaterials, owing to their exceptional electrical and optical properties, have garnered significant attention for applications in next-generation semiconductors, electronic switches, photodetectors, and other optoelectronic devices, demonstrating immense potential and research value. Typically, ZnO exists in a stable wurtzite structure, but when thinned to a 2D form, it transforms into a monolayer honeycomb structure reminiscent of graphene [18]. Compared with its 3D counterpart, 2D ZnO boasts a larger surface-to-volume ratio, enhancing its photoelectric properties and facilitating the formation of low-dimensional conductive channels. In recent years, both simulation and experimental studies on the doping and modification of g-ZnO have gradually progressed.

Tu’s [19] research delved into the elastic, piezoelectric, electronic, and optical properties of g-ZnO, analyzing phonon dispersion relations, elastic and piezoelectric constants, electronic band structures, and optical dielectric functions. Their research studies reveal that g-ZnO is significantly softer than graphene, uncovering the physical characteristics and potential applications as a wide-bandgap semiconductor. Peng et al. [20] conducted a thorough investigation into the mechanical properties of g-ZnO, comparing its stiffness, strength, Poisson’s ratio, and higher-order elastic constants with other 2D materials like g-BN. They found that second-order elastic constants increased monotonically with pressure, while Poisson’s ratio decreased. Other researchers systematically explored the effects of doping g-ZnO with metalloid elements (B, Si, Ge, As, Sb, and Te) [21] and non-metal elements (N and F) [22] on its geometric, electronic, and optical properties. These studies provided vital theoretical insights for designing optoelectronic devices and optical nanostructures based on g-ZnO. By precisely controlling dopant elements and concentrations, the electronic, magnetic, and optical properties of ZnO monolayers can be tailored to meet various application requirements. Further investigations delved into the influence of doping and defects on g-ZnO’s electronic structure, magnetism, optics, and photocatalytic performance, highlighting the potential of doping effects in nanoelectronics and spintronics. For instance, Cu doping combined with vacancies transformed g-ZnO’s energy band structure from direct to indirect, enabling bandgap tuning [23]. P-doped g-ZnO monolayers, regardless of the presence of oxygen or zinc vacancies, exhibited ferromagnetism attributed to spin polarization in P-3p orbitals and weakly bound O-2p electrons from Zn vacancies. With a Curie temperature above room temperature, this stability offers new possibilities for diluted magnetic semiconductors [24]. Moreover, studies have reported on the adsorption properties of various g-ZnO doping systems. For instance, B-, N-, and C-doped g-ZnO exhibit stronger chemical adsorption towards CO molecules, demonstrating spontaneous magnetization and variable magnetic properties during CO adsorption, suggesting potential applications in sensors or novel magnetic devices [25]. When Cr atoms adsorb on g-ZnO, they preferentially attach to the top of O atoms, transforming the Cr-ZnO monolayer into a metallic and magnetic material [26]. Alkali metals can stably adsorb on the center top of Zn-O hexagonal rings, reducing the bandgap of X/ZnO (X = Li, Na, K, Rb, or Cs) monolayers to zero, imparting metallic characteristics, significantly lowering the work function and ionization potential, and increasing the Fermi level, favoring electron emission [27]. Transition metal (TM) atoms (V, Cr, Mn, Fe, and Co) can modify g-ZnO through the adsorption or substitution of Zn, offering a promising method to tune the work function of g-ZnO over a wide energy range [28]. S. Chowdhury et al. [29] investigated the effects of biaxial strain (ε_xy_) on the formation energy, electronic structure, and optical absorption of Al-doped g-ZnO. Variations in ε_xy_ did not alter the direct-bandgap nature of g-ZnO. When ε_xy_ = +5%, the formation energy exhibited a linear increase, whereas for ε_xy_ > 5%, the formation energy demonstrated an inverse parabolic relationship with ε_xy_. Notably, Al-doped g-ZnO under a biaxial strain of ε_xy_ = +10% achieved the highest absorption within the visible-light range. Furthermore, Nazir et al. [30] delved into the structural evolution, electronic properties, and enhanced optical effects of g-ZnO under high-pressure conditions. Their research highlighted the unique characteristics of g-ZnO as a direct-bandgap semiconductor, with its bandgap significantly tunable to a wider range within a hydrostatic pressure range of 0 to 50 GPa. Further optical analysis unveiled the high sensitivity of the real part of the dielectric constant and refractive index to pressure changes, offering a novel perspective for manipulating the optical behavior of materials.

Despite the ongoing fervor in g-ZnO research, studies focusing on p-type g-ZnO leveraging its unique properties remain scarce. Given the myriad advantages of g-ZnO, this paper aims to achieve the p-type conversion of g-ZnO through elemental doping.

## 2. Theoretical Calculation Methodology

In this thesis, first principles calculations were conducted by using the CASTEP module of the Materials Studio software package (Accelrys Inc., Materials Studio version 2020.08, San Diego, CA, USA). Specifically, we leveraged density functional theory (DFT) and the ultra-soft pseudopotential framework to delve into the computational study [31]. With a focus on various doping systems of g-ZnO monolayer materials, we adopted the Perdew–Burke–Ernzerhof (PBE) [32] form of general gradient approximation (GGA) [33] as the core exchange–correlation functional to accurately simulate and optimize their geometric structures. We set a cutoff energy of 600 eV to adequately capture the electronic interactions. Additionally, a 2 × 2 × 1 k-point grid was employed in the Brillouin zone to optimize the balance between computational efficiency and precision. Furthermore, to prevent the interaction between periodic images, a vacuum layer with a thickness of 10 Å was set for g-ZnO along the *z*-axis direction.

## 3. Results and Discussion

### 3.1. Atomic Structure

In our computational model, we constructed a ZnO 4 × 4 × 1 supercell monolayer structure containing 32 atoms, where the undoped intrinsic g-ZnO monolayer is composed of Zn_16_O_16_, ensuring the representativeness and comprehensiveness of our calculations. After optimization, the lattice constant(*a*) of g-Zn_16_O_16_ was found to be 3.28 Å, which is in good agreement with the results of 3.27 Å reported in ref. [29] and 3.28 Å in ref. [34]. The Zn-O bond length in ref. [21] is 1.89 Å, while in ref. [35], it is 1.898 Å, both of which are close to our calculated result of 1.895 Å. Subsequently, through the combined doping of Li, Ag, and F atoms, we designed and implemented six distinct doping structures, g-Zn_15_LiO_16_, g-Zn_15_AgO_16_, g-Zn_14_LiAgO_16_, g-Zn_15_LiFO_15_, g-Zn_14_LiAgFO_15_, and g-Zn_14_Li_2_FO_15_, aiming to explore the impact of different doping strategies on the properties of the g-ZnO monolayer. To ensure the accuracy and reliability of our results, we precisely defined the valence electron configurations of each element: Zn-3d^10^4s^2^, O-2s^2^2p^4^, Li-1s^2^2s^1^, Ag-4d^10^5s^1^, and F-2s^2^2p^5^. Through this meticulously designed computational approach, we systematically investigated the characteristic changes in the electronic structure, band structure, and density of states of the pure g-ZnO monolayer and its various doping systems. In particular, we focused on elucidating the mechanisms governing the impact of single-acceptor doping, acceptor–donor co-doping, and other strategies centered on Li doping on the properties of the g-ZnO monolayer. Figure 1 visually presents the diagrams of g-ZnO monolayer doping systems, providing a clear visual foundation for understanding subsequent computational results.

The formation energy can indicate the degree of difficulty in doping g-Zn_16_O_16_ and the stability of the doped system. The formula for calculating the formation energy is
*E_f_* = *E*(*doped*) − *E*(*bulk*) + ∑*n_i_μ_i_* ,(1)
where *E_f_* is the formation energy, *E*(*doped*) is the total energy of the doped system, *E*(*bulk*) is the total energy of the pure system, *n_i_* is the number of atoms of element *i* transferred from the pure system to the doped system, and *μ_i_* is the chemical potential of element *i*. By adjusting the preparation environment, g-ZnO can be prepared under O-rich or Zn-rich conditions. Table 1 shows the formation energies of all doped systems. It can be seen from the table that the formation energies of all six doped systems under O-rich conditions are negative and smaller than those under Zn-rich conditions, indicating that the six doped systems of g-Zn_16_O_16_ are easier to dope and have stabler structures under O-rich conditions compared with Zn-rich conditions.

### 3.2. Electronic Properties

Figure 2 depicts the energy band structure diagrams of various doping systems from the *G* point to the *F* point interval. The blue line represents the conduction band minimum (CBM), the red line represents the valence band maximum (VBM), and the Fermi level is defined at 0 eV, as indicated by the dashed line. As seen in the figure, the band structure of pure graphene-like ZnO is similar to that of wurtzite-structured pure ZnO. When Li and Ag atoms substitute Zn atoms in the g-ZnO monolayer, the doping of acceptor elements results in the formation of shallow acceptor energy levels in g-Zn_15_LiO_16_ and g-Zn_15_AgO_16_. The energy band structure of g-Zn_14_LiAgO_16_ forms a deep acceptor energy level due to the co-doping of Li and Ag atoms in the g-ZnO monolayer, where the combined doping of two acceptor elements deepens the acceptor energy level. With Li and F elements co-doped into the g-ZnO monolayer, when the doping ratio equals to 1:1, the F atom, as a donor element, compensates for the shallow acceptor energy level formed by Li doping. Therefore, the energy band structure of g-Zn_15_LiFO_15_ does not form an acceptor energy level. Meanwhile, both g-Zn_14_LiAgFO_15_ and g-Zn_14_Li_2_FO_15_ exhibit shallow acceptor energy levels in their energy band structures, as shown in Figure 2. When two acceptor elements and one donor element are co-doped into the g-ZnO monolayer, the donor element compensates for the deep acceptor energy level formed by the co-doping of the two acceptor elements, resulting in the formation of shallow acceptor energy levels in the doping systems. Shallow acceptor energy levels facilitate electron transitions, thus promoting the formation of p-type semiconductors.

In Figure 3, the bandgaps and work functions of different doping systems are presented, with the left Y-axis representing the bandgap size and the right Y-axis indicating the work function size. The bandgap of g-Zn_16_O_16_ is only 1.68 eV, which is mainly attributed to underestimation by the GGA function in first principles calculations. However, this does not affect the comparison of different doping systems presented in this paper. When considering the energy band structure diagram in Figure 2, the doping of a single Li atom causes the CBM to shift more towards higher energy levels compared with the VBM, thereby increasing the forbidden energy band width. The doping of a single Ag atom shifts the CBM towards lower energy levels and the VBM towards higher energy levels, resulting in a narrower forbidden energy band width and a reduced bandgap. The bandgap of the Li-Ag co-doping structure is similar to that of g-Zn_16_O_16_. When Li-F are co-doped into the g-ZnO monolayer, the VBM remains almost unchanged, while the CBM shifts significantly towards higher energy levels, causing the bandgap of g-Zn_15_LiFO_15_ to be much larger than that of g-Zn_16_O_16_. The bandgap of g-Zn_14_Li_2_FO_15_ is approximately 0.3 eV larger than that of pure g-Zn_16_O_16_, while the bandgap of g-Zn_14_LiAgFO_15_ is approximately 0.3 eV smaller than that of pure g-Zn_16_O_16_. The size of the bandgap can reflect the ease of electron transition: the smaller the bandgap, the easier the electron transition and the better the electrical conductivity.

The work function (ϕ) of a semiconductor refers to the minimum energy required for electrons at the Fermi level (*E_f_*) in the semiconductor to transition to the vacuum level (*V_vac_*). The work function formula of the g-ZnO doping system can be estimated as
(2)$$\phi =V_{vac}-E_{f}\;.$$

The magnitude of the work function indicates the strength of electron binding in a metal. The larger the work function, the more difficult it is for electrons to leave the metal. The Y-axis on the right side of Figure 3 represents the magnitude of the work function. The work function of g-Zn_16_O_16_ is 4.876 eV. Compared with g-Zn_16_O_16_, the work function of g-Zn_15_LiO_16_ increases, while the work function of g-Zn_15_AgO_16_ decreases. This indicates that the doping of Li atoms makes it more difficult for electrons to escape from the semiconductor, which may be due to the change in the energy band structure of the semiconductor caused by Li atom doping, resulting in a decrease in the Fermi level. The doping of Ag atoms makes it easier for electrons to escape from the semiconductor, probably because Ag atom doping increases the Fermi level. The work function of g-Zn_14_LiAgO_16_ is 5.104 eV. Compared with g-Zn_15_LiO_16_, although the Ag atom is added, the work function still increases. This may be because the doping effect of the Li atom is more significant, leading to a further decrease in the Fermi level. The work functions of g-Zn_15_LiFO_15_ and g-Zn_14_Li_2_FO_15_ are 5.03 eV and 5.088 eV, respectively. Both systems contain the doping of Li atoms and F atoms. Compared with g-Zn_15_LiO_16_, the introduction of the F atom seems to have little effect on the work function, but the doping effect of the Li atom remains significant. The work function of g-Zn_14_LiAgFO_15_ is 4.641 eV. Compared with g-Zn_15_AgO_16_, although Li and F are added in a 1:1:1 ratio with Ag, the work function is still close to g-Zn_15_AgO_16_, indicating that the doping effect of Ag still dominates in this system. The lower work function facilitates electron injection and transport, which can enhance device performance in terms of conductivity and response time.

Figure 4 presents the total density of states (TDOS) and atomic-projected partial density of states (PDOS) for different doping systems of g-Zn_16_O_16_. Figure 4a shows the density of states (DOS) for g-Zn_15_LiO_16_, where one Zn atom is replaced by a Li atom. The posterior part of the valence band is primarily due to Zn3d orbitals, while the fore part of the valence band is mainly dominated by O2p orbitals. Li doping makes a minimal contribution to the valence and conduction bands. Near the Fermi level, hybridization occurs between Li1s2s electronic states and O2p electronic states, causing the Fermi level to enter the valence band. Figure 4b depicts the DOS for g-Zn_15_AgO_16_, where a Ag atom replaces a Zn atom. The posterior part of the valence band is primarily due to Zn3d orbitals, while the fore part of the valence band is dominated by both O2p and Ag4d orbitals. The Fermi level entering the valence band is primarily due to the hybridization between O2p electronic states and Ag4d electronic states. Figure 4c represents the DOS for g-Zn_14_LiAgO_16_. The contributions of various orbitals to the valence band are consistent with the conclusions from the previous two doping systems. Near the Fermi level, hybridization occurs among O2p electronic states, Ag4d electronic states, and Li1s2s electronic states. Figure 4d shows the DOS for g-Zn_14_Li_2_FO_15_, where two Zn atoms are replaced by Li atoms and one O atom is replaced by an F atom. Zn3d orbitals make a significant contribution to the posterior part of the valence band, with some contribution from F2p orbitals as well. The fore part of the valence band is still primarily dominated by O2p orbitals. Near the Fermi level, hybridization occurs primarily between O2p electronic states and Li1s2s electronic states, with minimal contribution from F. Figure 4e displays the DOS for g-Zn_14_LiAgFO_15_, where the posterior part of the valence band is due to both Zn3d and F2p orbitals, while the fore part of the valence band is dominated by O2p and Ag4d orbitals. Near the Fermi level, hybridization among O2p electronic states, Ag4d electronic states, and Li1s2s electronic states forms the characteristic DOS profile.

Table 2 presents the charge population of different doping systems of g-ZnO. From this, it can be concluded that in g-Zn_16_O_16_, the valence electrons of the Zn atoms are mainly distributed in the d-orbitals, with an average loss of 0.93 valence electrons, while the valence electrons of the O atoms are mainly distributed in the p-orbitals, gaining an average of 0.93 valence electrons, making pure g-ZnO non-conductive. In g-Zn_15_LiO_16_, the Zn atoms lose an average of 0.925 valence electrons, and the O atoms gain an average of 0.924 valence electrons, which is almost negligible compared with g-Zn_16_O_16_. Meanwhile, the Li atom loses 0.89 valence electrons, and g-Zn_15_LiO_16_ transfers a total of 0.981 valence electrons, with the number of lost valence electrons exceeding the number gained. In g-Zn_15_AgO_16_, the Zn atoms lose an average of 0.936 valence electrons, the O atoms gain an average of 0.923 valence electrons, and the Ag atom loses 0.68 valence electrons, resulting in an overall loss of 0.693 valence electrons. In g-Zn_14_LiAgO_16_, the Zn atoms lose an average of 0.922 valence electrons, the O atoms gain an average of 0.913 valence electrons, and the Li and Ag atoms lose 0.89 and 0.79 valence electrons respectively, leading to an overall loss of 1.689 valence electrons. In g-Zn_14_LiAgFO_15_, the Zn atoms lose an average of 0.913 valence electrons; the O atoms gain an average of 0.921 valence electrons; the Li and Ag atoms lose 0.9 and 0.72 valence electrons, respectively; and the F atoms gain 0.58 valence electrons, resulting in an overall loss of 1.032 valence electrons. In g-Zn_14_Li_2_FO_15_, the Zn atoms lose an average of 0.906 valence electrons, the O atoms gain an average of 0.924 valence electrons, the Li atoms lose 0.9 and 0.88 valence electrons, and the F atoms gain 0.58 valence electrons, leading to an overall loss of 0.282 valence electrons. In all the above doping systems, the number of lost valence electrons exceeds the number gained, forming hole-type conductivity and thus p-type ZnO.

Figure 5 illustrates the differential charge density diagram of p-type g-ZnO achieved through Li doping. The minimal overlap in charge density between Zn and O atoms signifies the covalent nature of the Zn-O bond. Upon closer inspection of Figure 5a–c, it becomes evident that when Li atoms replace the Zn atoms, the charge distribution between the Li atoms and the O atoms notably decreases. In Figure 5b, the introduction of Ag dopants leads to a slight fading in the charge distribution between the Ag atoms and the O atoms. Furthermore, in both Figure 5b,c, where F atoms substitute the O atoms, a diminution in the charge distribution between the Zn atoms and the F atoms is observed. This reduction in charge distribution for Li-O, Ag-O, and Zn-F bonds is directly correlated with the elongation of their respective bond lengths. Consequently, the substitution of Zn atoms with Li atoms or Ag atoms or the replacement of O atoms with F atoms not only weakens the covalent character of these bonds but also enhances their ionic nature.

### 3.3. Optical Properties

The dielectric function holds significant importance in optics and is typically expressed in a complex form as $\varepsilon \left(\omega \right)=\varepsilon_{1}\left(\omega \right)+i\varepsilon_{2}\left(\omega \right)$. The real part of the dielectric function, $\varepsilon_{1}\left(\omega \right)$, represents the linear response capability of a material to an electric field, which is essentially the degree of polarization under the action of an electric field, and is closely related to the refractive index of the material. The larger $\varepsilon_{1}\left(\omega \right)$ is, the stronger the material’s response to the electric field, indicating a higher degree of polarization and consequently a higher refractive index for light. The imaginary part of the dielectric function, $\;\varepsilon_{2}\left(\omega \right)$, is also a crucial parameter in assessing the optical properties of a material, directly related to its absorption coefficient. The larger $\varepsilon_{2}\left(\omega \right)$ is, the stronger the material’s ability to absorb light energy, resulting in greater energy loss. Together, the real and imaginary parts of the dielectric function determine the application performance of a material in optics, with optical parameters such as absorption coefficient *α*(*ω*) and refractive index *n*(*ω*) being all derivable from $\varepsilon_{1}\left(\omega \right)$ and $\varepsilon_{2}\left(\omega \right)$ by using the following formulas:(3)$$\alpha \left(\omega \right)=\sqrt{2}\omega {\left[\sqrt{\varepsilon_{1}^{2}\left(\omega \right)+\varepsilon_{2}^{2}\left(\omega \right)}-\varepsilon_{1}\left(\omega \right)\right]}^{\frac{1}{2}}\;.$$
(4)$$n\left(\omega \right)={\left[\sqrt{\varepsilon_{1}^{2}\left(\omega \right)+\varepsilon_{2}^{2}\left(\omega \right)}+\varepsilon_{1}\left(\omega \right)\right]}^{\frac{1}{2}}/\sqrt{2}\;.$$

Figure 6 presents the optical parameter curves of the g-ZnO doping systems. During simulation calculations, the underestimation of the bandgap by the GGA necessitated the application of a scissors operator to correct for this error in analyzing the optical properties. As shown in Figure 6a, the real part of the dielectric function, $\varepsilon_{1}\left(\omega \right)$, gradually decreases for each doping system until the wavelength exceeds 520 nm, at which point the $\varepsilon_{1}\left(\omega \right)$ of g-Zn_14_LiAgFO_15_ becomes larger than that of pure g-ZnO. The variation curves for single-Li-atom doping and single-Ag-atom doping are quite similar, with g-Zn_15_LiO_16_ exhibiting a smaller $\varepsilon_{1}\left(\omega \right)$ than g-Zn_15_AgO_16_ after 400 nm. Within the visible-light wavelength range of 380 nm to 780 nm, the four doping systems (g-Zn_15_LiO_16_, g-Zn_15_AgO_16_, g-Zn_14_LiAgO_16_, and g-Zn_14_Li_2_FO_15_) all exhibit a trough, where particularly notable is the negative value of $\varepsilon_{1}\left(\omega \right)$ for g-Zn_14_Li_2_FO_15_ around 440 nm. In Figure 6b, once the wavelength enters the visible-light range, the imaginary part of the dielectric function $\varepsilon_{2}\left(\omega \right)$ is greater than that of pure g-ZnO for all doping systems, except for g-Zn_15_LiFO_15_, whose $\varepsilon_{2}\left(\omega \right)$ remains smaller than that of pure g-ZnO throughout. Furthermore, these four doping systems—g-Zn_15_LiO_16_, g-Zn_15_AgO_16_, g-Zn_14_LiAgO_16_, and g-Zn_14_Li_2_FO_15_—display pronounced peaks in $\varepsilon_{2}\left(\omega \right)$ within the visible-light region, with g-Zn_14_Li_2_FO_15_ exhibiting a peak value approximately twice that of the other three systems.

Figure 6c displays the absorption curves of the g-ZnO doping systems. Among them, g-Zn_15_LiO_16_, g-Zn_15_AgO_16_, g-Zn_14_LiAgO_16_, and g-Zn_14_Li_2_FO_15_ exhibit broad and prominent absorption peaks in the visible-light region. Notably, the peak of g-Zn_14_Li_2_FO_15_ is significantly enhanced, aligning well with the trend observed in the imaginary part of the dielectric function $\varepsilon_{2}\left(\omega \right)$, confirming the direct correlation between $\varepsilon_{2}\left(\omega \right)$ and the material’s absorption coefficient mentioned earlier. High absorption implies that more light energy can be absorbed by the material and converted into other forms, potentially making it a candidate for optical storage materials. Figure 6d illustrates the reflectance behavior. Compared with g-Zn_16_O_16_, g-Zn_14_LiAgFO_15_ shows a slight increase in reflectance in the visible-light region, while g-Zn_15_LiO_16_, g-Zn_15_AgO_16_, g-Zn_14_LiAgO_16_, and g-Zn_14_Li_2_FO_15_ experience a sharp enhancement, particularly g-Zn_14_Li_2_FO_15_, whose reflectance peak around 500 nm exceeds that of the other three by more than 0.1. Figure 6e and Figure 6f depict the refractive index and energy loss curves, respectively, of the g-ZnO doping systems. Figure 6e closely resembles the variation curve of the real part of the dielectric function $\varepsilon_{1}\left(\omega \right)$, shown in Figure 6a, indicating the accuracy of the calculations. The absorption rate of a semiconductor material directly determines the energy loss of light within the material. Higher absorption rates signify more light energy being absorbed and converted into other forms, leading to greater energy loss. In Figure 6f, the energy loss spectrum describes the distribution of energy loss when light passes through matter; g-Zn_14_Li_2_FO_15_ displays a sharp and narrow peak near 450 nm, which corresponds to its high absorption rate in the visible-light region.

## 4. Conclusions

This paper investigates the p-type transition of graphene-like zinc oxide (g-ZnO) through elemental doping by using first principles calculations. It reveals and analyzes the electronic structures and optical properties of various doping g-ZnO systems. Specifically, the effects of single doping, co-doping, and donor–acceptor co-doping with elements such as Li, Ag, and F on g-ZnO are studied in detail. The research findings are as follows:Both Li and Ag, as acceptor elements, can form shallow acceptor levels when doped individually into g-ZnO. However, in the case of g-Zn_15_LiFO_15_, the presence of the donor element F compensates for the acceptor effect, resulting in the absence of acceptor levels. Conversely, g-Zn_14_LiAgFO_15_ and g-Zn_14_Li_2_FO_15_ achieve shallow acceptor levels by compensating for the deep acceptor levels formed by the double-acceptor elements (Li and Ag) with the donor element F. Single-acceptor doping and acceptor–donor co-doping can render g-ZnO a p-type semiconductor.The bandgap and work function of the doped systems vary depending on the doping elements and methods. Notably, the co-doping strategy involving double-acceptor elements and donor elements exhibits advantages in reducing the bandgap and enhancing carrier mobility. The reduction in the bandgap makes the electron transition between the valence band and the conduction band easier, enhancing the conductivity of the doped system.The optical properties of the doped systems are investigated through dielectric functions, optical absorption spectra, reflection spectra, refractive indices, and energy loss functions. Among them, g-Zn_14_Li_2_FO_15_ stands out in the visible-light region compared with other doped systems, providing theoretical support for the application of p-type g-ZnO in optoelectronic fields, such as the hole transport layer of solar cells.

## Figures and Tables

**Figure 1 nanomaterials-14-01863-f001:**
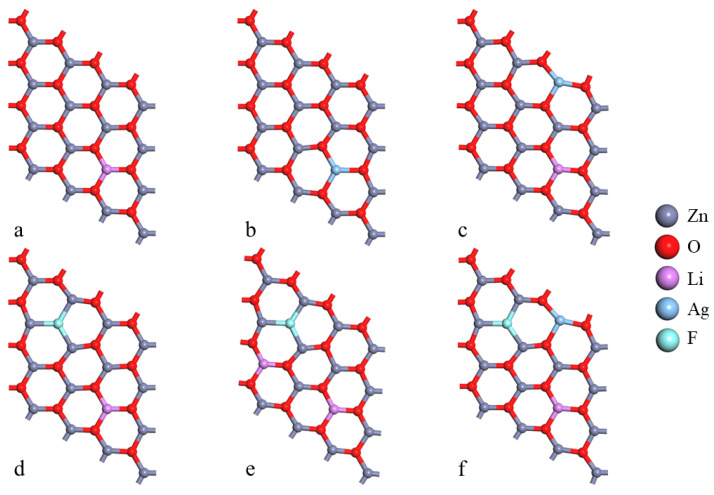
The model of (4 × 4 × 1) g-ZnO monolayer (**a**) g-Zn_15_LiO_16_, (**b**) g-Zn_15_AgO_16_, (**c**) g-Zn_14_LiAgO_16_, (**d**) g-Zn_15_LiFO_15_, (**e**) g-Zn_14_Li_2_FO_15_, and (**f**) g-Zn_14_LiAgFO_15_.

**Figure 2 nanomaterials-14-01863-f002:**
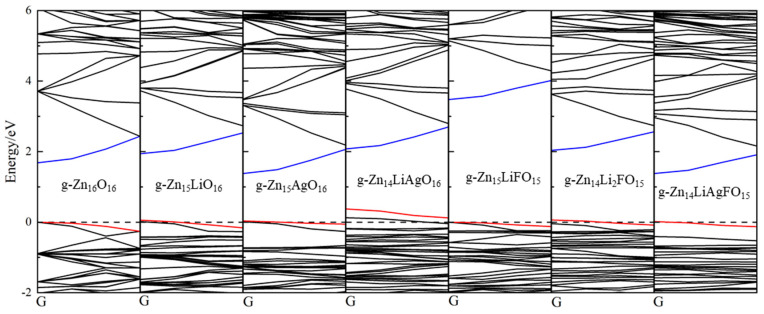
The energy band structure diagrams of various doping systems from the *G* point to the *F* point interval.

**Figure 3 nanomaterials-14-01863-f003:**
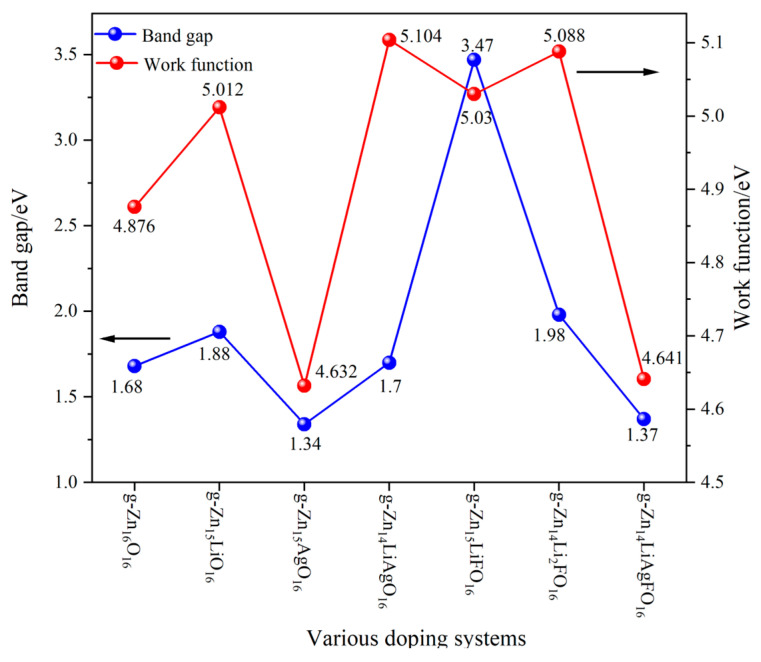
The bandgaps and work functions of different doping systems.

**Figure 4 nanomaterials-14-01863-f004:**
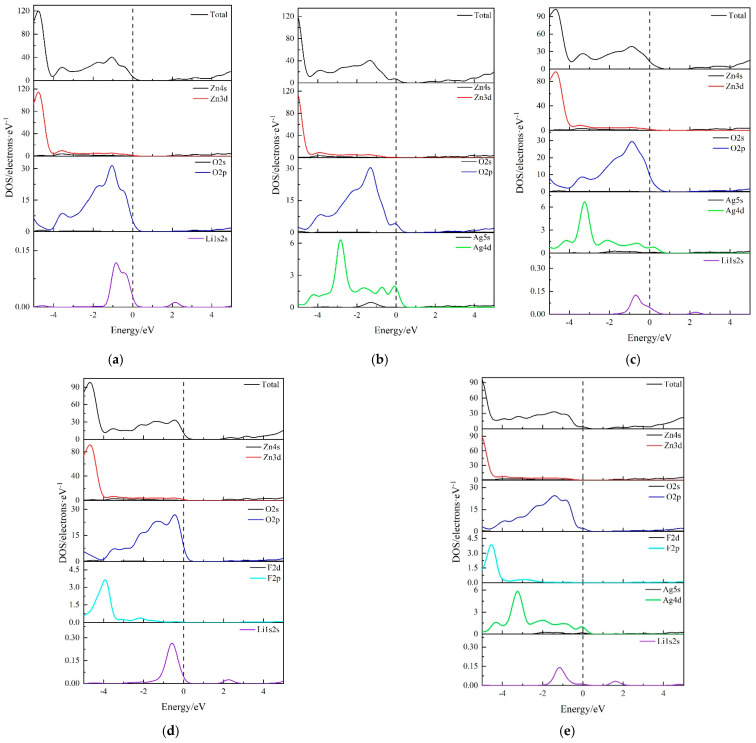
The TDOS and PDOS of (**a**) g-Zn_15_LiO_16_, (**b**) g-Zn_15_AgO_16_, (**c**) g-Zn_14_LiAgO_16_, (**d**) g-Zn_14_Li_2_FO_15_, and (**e**) g-Zn_14_LiAgFO_15_.

**Figure 5 nanomaterials-14-01863-f005:**
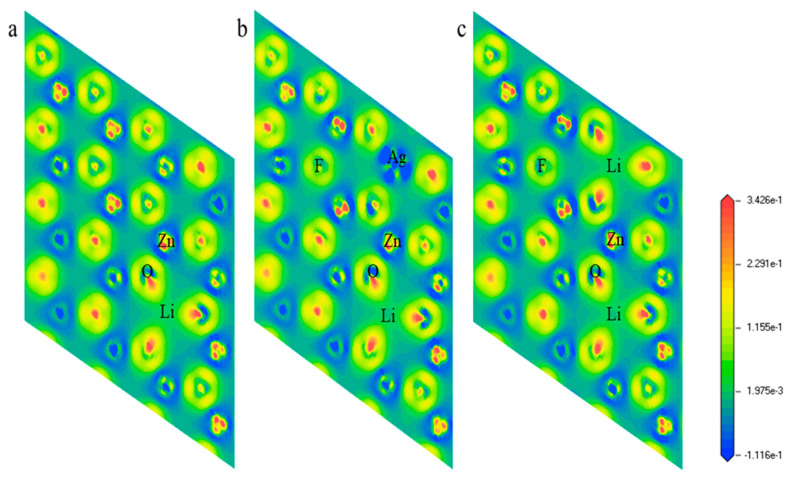
The differential charge density diagram of (**a**) g-Zn_15_LiO_16_, (**b**) g-Zn_14_Li_2_FO_15_, and (**c**) g-Zn_14_LiAgFO_15_.

**Figure 6 nanomaterials-14-01863-f006:**
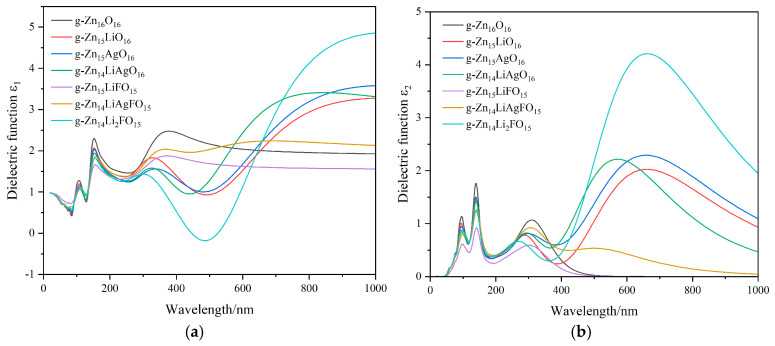
(**a**) The real part of the dielectric function $\varepsilon_{1}\left(\omega \right)$. (**b**) The imaginary part of the dielectric function $\varepsilon_{2}\left(\omega \right)$. (**c**) The optical properties of the absorption coefficient *α*(*ω*). (**d**) The optical properties of reflectance R(*ω*). (**e**) The optical properties of the refractive index *n*(*ω*). (**f**) The optical properties of the loss function L(*ω*).

**Table 1 nanomaterials-14-01863-t001:** The formation energies of all doped systems.

Models	*E_f_*/eV	
	O-Rich	Zn-Rich
g-Zn_15_LiO_16_	−3.3	0.07
g-Zn_15_AgO_16_	−1.657	1.713
g-Zn_14_LiAgO_16_	−5.41	1.33
g-Zn_15_LiFO_15_	−7.04	−7.04
g-Zn_14_LiAgFO_15_	−8.77	−5.4
g-Zn_14_Li_2_FO_15_	−10.28	−6.91

**Table 2 nanomaterials-14-01863-t002:** The charge population of different doping systems of g-ZnO.

Models	Element	s	p	d	Total Charge (e)	Net Charge (e)
g-Zn_16_O_16_	Zn	0.61	0.5	9.96	11.07	0.93
	O	1.86	5.07	0	6.93	−0.93
g-Zn_15_LiO_16_	Zn	0.616	0.514	9.946	11.075	0.925
	O	1.868	5.056	0	6.924	−0.924
	Li	2	0.11	0	2.11	0.89
g-Zn_15_AgO_16_	Zn	0.612	0.496	9.954	11.064	0.936
	O	1.864	5.059	0	6.923	−0.923
	Ag	0.51	0.14	9.68	10.32	0.68
g-Zn_14_LiAgO_16_	Zn	0.616	0.519	9.944	11.078	0.922
	O	1.869	5.043	0	6.913	−0.913
	Li	2.01	0.11	0	2.11	0.89
	Ag	0.51	0.14	9.56	10.21	0.79
g-Zn_14_LiAgFO_15_	Zn	0.628	0.508	9.951	11.087	0.913
	O	1.867	5.054	0	6.921	−0.921
	Li	2	0.09	0	2.1	0.9
	Ag	0.51	0.14	9.62	10.28	0.72
	F	1.96	5.62	0	7.58	−0.58
g-Zn_14_Li_2_FO_15_	Zn	0.631	0.521	9.941	11.094	0.906
	O	1.87	5.055	0	6.924	−0.924
	Li	2.015	0.115	0	2.12	0.88
	F	1.96	5.62	0	7.58	−0.58

## Data Availability

The raw/processed data required to reproduce these findings cannot be shared at this time, as the data also form part of an ongoing study.

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
