# Peer review of "First Principles Study of p-Type Transition and Enhanced Optoelectronic Properties of g-ZnO Based on Diverse Doping Strategies"

_nanomaterials, 2024, doi:10.3390/nano14231863_

Round 1

Reviewer 1 Report

Comments and Suggestions for Authors

Dr. Kaiqi Bao and coworkers report a theoretical approach to the doped graphene-like ZnO from the electronic and optical properties viewpoint. The manuscript may contain valuable information for researchers in the field of oxide semiconductors and low-dimensional materials. However, some content is not friendly to the readers.

1) The reader cannot assess the structural stability of the proposed models. The authors are advised to show the structural stability of all models from an energetic viewpoint quantitatively.

2) In addition to the above-mentioned structural stability, the authors are also advised to show the effect of the positions of the dopant. Especially, relative positional relations of dopants in the case of co-doping.

3) The authors refer to the data of the pure Wurtzite ZnO for comparison. In this case, the authors are advised to show the those of the pure Wurtzite ZnO. All figures and tables should include data on the pure Wurtzite ZnO. 

4) In Figure 2, the relative relations of VBM (blue line) and Fermi level (dashed line) are not clear. The authors are advised to show the enlarged figures near the Fermi level. In addition, the line denotes the Fermi level is not a dotted line but a dashed line.

5) In Figure 5, the correspondence of models and data is not clear. The authors are advised to clearly show these correspondences.

For these reasons, the authors are advised to revise the manuscript. 

Author Response

Comments 1: The reader cannot assess the structural stability of the proposed models. The authors are advised to show the structural stability of all models from an energetic viewpoint quantitatively.

Response 1:

Thank you for pointing this out. We agree with this comments. We have incorporated the formation energy into the article to explain the ease of formation of the doped system and the stability of its structure. Specifically, the content related to formation energy has been added both in the text between lines 140 and 152 and in Table 1.

Comments 2:  In addition to the above-mentioned structural stability, the authors are also advised to show the effect of the positions of the dopant. Especially, relative positional relations of dopants in the case of co-doping.

Response 2:

Thank you for this suggestion. Regarding your suggestion to increase the calculation of doping positions, due to the nature of the doping model, many doping positions fail to converge during simulation calculations. Furthermore, the primary focus of this paper is to discuss the impact of different doping elements on performance, and doping at different positions has little influence on electrical and optical properties.

Comments 3: The authors refer to the data of the pure Wurtzite ZnO for comparison. In this case, the authors are advised to show the those of the pure Wurtzite ZnO. All figures and tables should include data on the pure Wurtzite ZnO.

Response 3:

Thank you for pointing this out. In the Results and Discussion section of this paper when analyzing the simulation results, the wurtzite ZnO structure was not mentioned. The introduction of wurtzite ZnO in the Introduction section was merely to introduce the topic. Therefore, in response to your suggestion, this section has not been supplemented with that information.

Comments 4:  In Figure 2, the relative relations of VBM (blue line) and Fermi level (dashed line) are not clear. The authors are advised to show the enlarged figures near the Fermi level. In addition, the line denotes the Fermi level is not a dotted line but a dashed line.

Response 4:

Agree. We have, accordingly, changed. In this paper, the bottom of the conduction band in Figure 2 has been changed to a blue line, and the top of the valence band has been changed to a red line. This should make it clearer to see the relationship between the top of the valence band and the Fermi level. Additionally, the dotted lines in the text have been changed to dashed lines.

Comments 5: In Figure 5, the correspondence of models and data is not clear. The authors are advised to clearly show these correspondences.

Response 5:

We agree with this comments and have made the necessary changes accordingly. In Figure 5, the numerical values indicating the degree of electron gain or loss have been marked, and the key elemental positions have also been indicated in the diagram.

Reviewer 2 Report

Comments and Suggestions for Authors

Manuscript ID: nanomaterials-3286930

Comprehensive Comments:

The research article titled "Investigation of p-Type Transition and Enhanced Optoelectronic Properties of g-ZnO via Diverse Doping Strategies" presents valuable insights into the p-type doping mechanisms of graphene-like zinc oxide (g-ZnO) through computational methods. The use of first-principles calculations to analyze various doping strategies significantly advances the understanding of the electronic and optoelectronic properties of g-ZnO, a material with the promise for future applications in optoelectronic devices.

The methodology utilized, primarily based on first-principles computational techniques using density functional theory (DFT), is appropriate for the exploration of the intricate details of electronic structures and optical properties for the low-dimensional materials such as g-ZnO. While the analysis and discussion of results could benefit from a deeper theoretical interpretation, they do provide sufficient logical associations that relate observations of electronic structures and optical properties to the doping strategies employed. This would lay a considerable foundation for the future exploration of experimental verifications.

Suggestions for Major Revision:

1. Expand Discussion on Results: Incorporate a more in-depth analysis regarding the electronic structure results, particularly how the changes in band gap and work function relate to potential device applications. Discussing the implications of these results in the context of existing literature can also offer additional value.

2. Figures and Legends: Ensure that all figures and their legends are clear and detailed. Some figures, particularly those depicting band structures and densities of states, should be labeled with numerical values or clearer annotations, allowing the reader to easily extract the critical information without excessive cross-referencing in the text.

3. Revisit Optical Properties Analysis: While the results regarding optical properties are interesting, the section could be made more comprehensive by including a discussion on the implications of these properties for photonic applications. A comparative analysis of the optical performance against known materials in literature might also add significance.

4. Address Computational Limitations: Acknowledge any limitations of the computational study, such as the potential inaccuracies in predicted band gaps or dielectric functions due to the chosen theoretical approach. Discussing how these might affect the conclusions would enhance the paper's transparency.

5. Conclusion Enhancement: The conclusions should be revised to more clearly outline the broader impact of the research findings and how they contribute to the field of optoelectronic materials. Emphasize the significance of achieving p-type conductance in g-ZnO and its potential applications.

6. Physical Note: Line 114, “a vacuum layer with a thickness of 10” is 10Å or 10 number of layer thickness/unit cell dimension?

7. Paper Title Vague: “Investigation” should be clarified by substituting it with “First-principles study”.

8. Non-standard Phrases: Section titles “2. Computational method models”, ”3.1. Models” and “3.2. Electrical properties” should be amended to “Theoretical calculation methodology”, “Atom structure” and “Electronic properties” (or “Electronic structures”) respectively; The “charge distribution” in the table 1 caption should be “charge population” as correct; The “band” arising in figure captions and main texts should be remedied to “energy band”; “partial density of states” in line 202 should be emphasized as “atomic-projected partial density of states” as adequate.

9. Software Claim: Please give out which version of Materials Studio utilized in this study, such as “Materials Studio 2020 package (Accelrys Inc., Materials Studio version 2020.08, San Diego, CA, USA)”.

10. Redundant: Equations 1-5, and corresponding paragraphs indicating them, which are all such useless and general, could be deleted as better.

11. Physical Concept Improvement: Line 338-340, it should be modified as “It reveals and analyzes the electronic structures and optical properties of various doping g-ZnO systems”.

Conclusive Recommendation:

This article presents significant findings that contribute to the understanding of p-type doping in g-ZnO, which has the potential to drive further exploration and development in the field of optoelectronic materials. Therefore, after addressing above-mentioned revise points, I recommend a major revision on this article before considering accept it for publication in Nanomaterials.

Comments on the Quality of English Language

The English could be improved to more clearly express the research.

Author Response

Comments 1:  Expand Discussion on Results: Incorporate a more in-depth analysis regarding the electronic structure results, particularly how the changes in band gap and work function relate to potential device applications. Discussing the implications of these results in the context of existing literature can also offer additional value.

Response 1:

Thank you for this proposal. We have incorporated the following content into lines 192-194 and 217-219 respectively: the size of the band gap can reflect the ease of electron transition: the smaller the band gap, the easier the electron transition and the better the electrical conductivity.

The lower work function facilitates easier electron injection and transport, which can enhance device performance in terms of conductivity and response time.

Comments 2:  Figures and Legends: Ensure that all figures and their legends are clear and detailed. Some figures, particularly those depicting band structures and densities of states, should be labeled with numerical values or clearer annotations, allowing the reader to easily extract the critical information without excessive cross-referencing in the text.

Response 2:

Thanks for your advice. We agree with this viewpoint and have made the necessary modifications accordingly. In Figure 3, specific numerical values for each point have been labeled, and excessive explanations in the text have been removed.

Comments 3: Revisit Optical Properties Analysis: While the results regarding optical properties are interesting, the section could be made more comprehensive by including a discussion on the implications of these properties for photonic applications. A comparative analysis of the optical performance against known materials in literature might also add significance.

Response 3:

Thank you for pointing this out. Supplements regarding the application fields of optics in relation to the optical properties have been added in lines 329-331 and 341-342 respectively. Follow as:

High absorption implies that more light energy can be absorbed by the material and converted into other forms, potentially making it a candidate for optical storage materials.

The energy loss spectrum describes the distribution of energy loss when light passes through matter.

Comments 4:  Address Computational Limitations: Acknowledge any limitations of the computational study, such as the potential inaccuracies in predicted band gaps or dielectric functions due to the chosen theoretical approach. Discussing how these might affect the conclusions would enhance the paper's transparency.

Response 4:

Agree. We have, accordingly, done. The supplement regarding the impact of computational limitations on the band gap has been added in lines 179-181:

The band gap of g-Zn16O16 is only 1.68 eV, which is mainly attributed to the underestimation by the GGA function in first-principles calculations. However, this does not affect the comparison of different doping systems presented in this paper.

Comments 5:  Conclusion Enhancement: The conclusions should be revised to more clearly outline the broader impact of the research findings and how they contribute to the field of optoelectronic materials. Emphasize the significance of achieving p-type conductance in g-ZnO and its potential applications.

Response 5:

Thanks for your proposal. For the in-depth additions to the conclusion section, the following supplements have been made in lines 366-367, 371-373, and 377-378 respectively:

Single acceptor doping and acceptor-donor co-doping can render g-ZnO a p-type semiconductor.

The reduction of the band gap makes the electron transition between the valence band and the conduction band easier, enhancing the conductivity of the doped system.

Providing theoretical support for the application of p-type g-ZnO in optoelectronic fields such as the hole transport layer of solar cells.

Comments 6: Physical Note: Line 114, “a vacuum layer with a thickness of 10” is 10Å or 10 number of layer thickness/unit cell dimension?

Response 6:

Thank you for your thorough review of the article and for raising this question. It has been corrected in line 114: a vacuum layer with a thickness of 10Å.

Comments 7: Paper Title Vague: “Investigation” should be clarified by substituting it with “First-principles study”.

Response 7:

Thank you very much for putting forward such valuable proposal. The article title has been changed to “First-principles study of p-Type transition and enhanced optoelectronic properties of g-ZnO via diverse doping strategies”

Comments 8:  Non-standard Phrases: Section titles “2. Computational method models”, ”3.1. Models” and “3.2. Electrical properties” should be amended to “Theoretical calculation methodology”, “Atom structure” and “Electronic properties” (or “Electronic structures”) respectively; The “charge distribution” in the table 1 caption should be “charge population” as correct; The “band” arising in figure captions and main texts should be remedied to “energy band”; “partial density of states” in line 202 should be emphasized as “atomic-projected partial density of states” as adequate.

Response 8:

Thank you for such a detailed revision. We have made all the corrections based on your suggestions for the subheadings. “2. Computational method models”, ”3.1. Models” and “3.2. Electrical properties” should be amended to “Theoretical calculation methodology”, “Atom structure” and “Electronic properties” respectively.

Due to the addition of a new table above, the original Table 1 has been changed to Table 2, and its caption has been revised to “Table 2. The charge population of different doping systems of g-ZnO.”

In the text, most instances of "band" have been changed to "energy band".

Due to the addition of content above, ' partial density of states ' has been changed to ' atomic-projected partial density of states’ in lines 222-223.

Comments 9: Software Claim: Please give out which version of Materials Studio utilized in this study, such as “Materials Studio 2020 package (Accelrys Inc., Materials Studio version 2020.08, San Diego, CA, USA)”.

Response 9:

Thank you for pointing this out. We have implemented changes based on your suggestions. The calculations in this paper utilized the CASTEP software package, as explained in line 105. The explanation is as follows: In this thesis, first-principles calculations were conducted using the CASTEP module.

Comments 10:  Redundant: Equations 1-5, and corresponding paragraphs indicating them, which are all such useless and general, could be deleted as better.

Response 10:

We agree with this comments and have made the necessary changes accordingly. Some equations have been removed, but not all, as removing all of them would have made the context feel disconnected. We have removed the formulas for reflectance R(ω) and energy loss spectrum L(ω).

Comments 11:  Physical Concept Improvement: Line 338-340, it should be modified as “It reveals and analyzes the electronic structures and optical properties of various doping g-ZnO systems”.

Response 11:

We sincerely thank you for your revision. We agree with this comment and have made the necessary changes accordingly. Due to the addition of content above, the original content in lines 338-340 has been moved down to lines 357-358, and the content has been revised to: It reveals and analyzes the electronic structures and optical properties of various doping g-ZnO systems.

We take this opportunity to once again express my sincere gratitude to you for your insightful and valuable suggestions. They have played a significant role in enhancing the quality of my paper. We have attached the revised version of the manuscript with changes marked where possible for easy comparison. If you require any further clarification or additional information, please do not hesitate to contact me at any time. We are more than willing to discuss any issues further.

Thank you once again for your invaluable contribution to my work. I look forward to the next step in the publication process.

Round 2

Reviewer 2 Report

Comments and Suggestions for Authors

The author has basically made corresponding revisions in the revised manuscript according to the reviewers suggestions, almost completely. Nevertheless, some details should be improved further such as follows:

1. Line 105, should make it clear which software is utilized in this study, as “using the CASTEP module of Materials Studio software package ---”. 

2. Throughout all the text and figure caption or table content, every symbol of physical quantity (note, not the physical unit of a physical quantity) should be italic, such as those should be revised in the paragraphs after Figure 1 and in Table 1.

3. Line 143 should be written in the top case.

Therefore, it is recommended for acceptance of this paper at more revised version to be published on Nanomaterials.

Author Response

Comments 1: Line 105, should make it clear which software is utilized in this study, as “using the CASTEP module of Materials Studio software package ---”.

Response 1:

Thank you for pointing this out. We agree with this comments. We have added your advice in line 105.

Comments 2: Throughout all the text and figure caption or table content, every symbol of physical quantity (note, not the physical unit of a physical quantity) should be italic, such as those should be revised in the paragraphs after Figure 1 and in Table 1.

Response 2:

Thank you very much for putting forward such valuable proposal. We have already changed the last paragraph of Figure 1 and the physical quantities in Table 1 to italic font, as well as corrected the physical quantities in other parts of the text.

Comments 3: Line 143 should be written in the top case.

Response 3:

Thank you for this proposal. We have capitalized the first letter of 'where' in line 143.